# A Study on Intentional-Value-Substitution Training
# for Regression with Incomplete Information

Takuya Fukushima [* 1]  Tomoharu Nakashima [* 1]  Taku Hasegawa [2]  Vicenç Torra [3]

## Abstract

This paper focuses on a method to train a regression model from incomplete input values. It is assumed in this paper that there are no missing values in a training data set while missing values exist during a prediction phase using the trained model. Under this assumption, Intentional-Value-Substitution (IVS) training is proposed to obtain a machine learning model that makes the prediction error as minimum as possible. Through a mathematical analysis, it is shown that there are some meaningful substitution values in the IVS training for the model. It is shown through a series of computational experiments that the substitution values estimated by the extended mathematical analysis help the models predict outputs for inputs with missing values even though there is more than one missing value.

## 1. Introduction

An ideal situation in terms of regression in general is that each data point is complete without any missing values as well as the training dataset is large enough to build an accurate model. However, this is a rare case in real-world problems. For example in medical diagnosis, some measurements might not be available due to the failure in the measuring equipment or patient's personal reasons.

There are several ways to overcome the issue of handling missing values (Baraldi & Enders, 2010). One way is to impute a missing value by a certain value (e.g., zero, the average value of feature values, or the output of an imputa-

tion model constructed from the training dataset). Another way is to construct a model without those features that include missing values. Some papers presented how to handle the incomplete data with missing values by using statistical modeling. Methods for the parameter estimation were also proposed in (Little & Rubin, 1986). Furthermore, the ways to handle missing values have been discussed as multiple imputation (Rubin, 1989) and maximum likelihood estimation (Schafer & Graham, 2002). Tresp et al.(Tresp et al., 1993) provided a way to incorporate missing or uncertain values during training of neural networks and showed that heuristic ways could be harmful in the training. Acock (Acock, 2005) discussed substitution strategies for missing values. He mentioned that non-optimum strategies for missing values could produce biased estimates, distorted statistical power, and invalid conclusions.

It should be noted that the above-mentioned methods consider the case where both training and test datasets have the missing values. On the other hand, we consider the case where there are missing values only in the test dataset and the training dataset is complete without any missing values. This situation happens in many real-world problems. For example, in emergency medical care and sports, a good amount of information is available in the training and learning phase, but in the practical situation (i.e., in the test phase in the context of machine learning), one must decide in a short time with a limited amount of information.

Hasegawa et al.(Hasegawa et al., 2019) proposed a method for training a data-driven model for the case where there are missing values only in the test data. In this paper, we refer to this method as Intentional-Value-Substitution (IVS) training. IVS training substitutes a non-missing value in the training dataset with some value. In other words, this method models the target function using a modified training dataset where some feature values are substituted with a certain value even though no missing values are contained in the datasets. Hasegawa et al.(Hasegawa et al., 2019) investigated the effectiveness of IVS training, and Fukushima et al.(Fukushima et al., 2019) proposed a method to estimate an appropriate value for value-substitution in the case of two-dimensional problems.

In this paper, we extend the previous estimation method

---

[*]Equal contribution  [1]Osaka Prefecture University, Osaka, Japan [2]Osaka Prefecture University, Osaka, Japan (Current affiliation: NTT Media Intelligence Laboratory, NTT Corporation, Tokyo, Japan) [3]Hamilton Institute, Maynooth University, Maynooth, Ireland. Correspondence to: Takuya Fukushima <takuya.fukushima@kis.osakafu-u.ac.jp>, Tomoharu Nakashima <tomoharu.nakashima@kis.osakafu-u.ac.jp>.

*Presented at the first Workshop on the Art of Learning with Missing Values (Artemiss) hosted by the $37^{th}$ International Conference on Machine Learning (ICML).* Copyright 2020 by the author(s).

to three-dimensional problems. We assume that the value-missing happens at the second and last dimensions with a certain probability, not at the first dimension. Under these assumptions, we propose the method for estimating the optimal substitution values.

## 2. Intentional-Value-Substitution (IVS) Training

In this section, we introduce the procedure of the IVS training for obtaining a robust machine learning model against missing values. Note that no missing values exist in a training dataset, while a test dataset contains missing values. On the other hand, we assume that we know in advance which features will contain missing values in a test dataset.

For the sake of simplicity, this paper define $\boldsymbol{x} = (x_1, x_2, \ldots, x_n)$ as an $n$-dimensional input vector drawn from the training dataset. Furthermore, we suppose that the $i$-th feature $x_i$ is possibly missing at the test dataset. Under such a situation, the procedure of IVS training is shown in the following three steps:

Step 1: Draw an input vector $\boldsymbol{x}$ with its associated target value from the training dataset.

Step 2: With a pre-specified probability, substitute $x_i$ for a certain value.

Step 3: Train a prediction model with the modified input vector and the target value.

We can easily expand the above procedure for mini-batch training by iterating the process as many as the number of input vectors in the batch set.

The training phase in the IVS training needs to consider which value is used for substitution. The setting is used in Step 2 of the above procedure.

Through a mathematical analysis, the optimal value that can minimize the expected error between the prediction of the model and the target value is obtained as follows:

$$\psi'_{D_{X_{\mathrm{mis}}}}(\boldsymbol{x}_{\mathrm{obs}})$$
$$= \underset{\boldsymbol{x}'_{\mathrm{mis}}}{\arg\min} \{ \int_{D_{X_{\mathrm{mis}}}} p(\boldsymbol{x}_{\mathrm{mis}}|\boldsymbol{x}_{\mathrm{obs}}) f(\boldsymbol{x}_{\mathrm{obs}}, \boldsymbol{x}_{\mathrm{mis}}) \, dX_{\mathrm{mis}}$$
$$- f(\boldsymbol{x}_{\mathrm{obs}}, \boldsymbol{x}'_{\mathrm{mis}}) \}^2, \tag{1}$$

where $\boldsymbol{x}_{\mathrm{obs}}$ is the value that never be missing even a test phase. On the other hand, $\boldsymbol{x}_{\mathrm{mis}}$ are possibly missing only in the test phase. In Eq. (1), some conditions are assumed(e.g., the target function and where the missing is possibly to occur are known, a prediction model has a sufficient accuracy in approximating the target function). The more detail explanation are in Appendix. A.

## 3. Estimation of Optimal Substitution Values without the Target Function

In the previous section, we obtained the function $\psi'(\cdot)$ by assuming that we know the target function $f$ beforehand. However, of course, the target function $f$ is unknown in many problem settings. On the other hand, as shown in the previous section, it is clear that the optimal substitution value has an important meaning in an imputation of missing values and IVS training.

For this problem, Fukushima et al.(Fukushima et al., 2019) proposed a method to estimate the optimal value without the target function in problem settings where features only on a single dimension are missing. In general, however, the value missing would happen simultaneously in practical problems that have more than two-dimensionality.

Therefore, in this section, we propose a method that can estimate the optimal substitution values even though multiple missing would happen.

### 3.1. Single Missing

First of all, we introduce a method that can estimate the optimal value of single-missing problems proposed in (Fukushima et al., 2019). For simplicity, it is assumed that the dimensionality of the problem is three. The method consists of the following four steps to calculate the function $\psi'(\cdot)$ to estimate optimal substitution values. In the following explanation, it is assumed that the missing occurs only at the third dimension.

The method is described as pseudo-code in Algorithm 1. Note that the method can only apply to problem settings where feature-missing would happen just on a particular dimension regardless of the number of dimensionalities.

### 3.2. Multiple Missing

In this paper, we extend the method mentioned in Subsec. 3.1 to be able to treat multiple missing. For simplicity, we assume that the dimensionality is set as three similarly, and the second and third elements might be missing. Therefore, we discuss the case where the features are missing on the second and third dimensions simultaneously. When each random variable is independent, Eq. (1) should be satisfied in all missing features, that is,

$$\psi'_2 = \underset{x'_2}{\arg\min} \{ \int_{-\infty}^{\infty} p(x_2|x_1, x_3) \tag{2}$$
$$f(x_1, x_2, x_3) dx_2 - f(x_1, x'_2, x_3) \}^2,$$

$$\psi'_3 = \underset{x'_3}{\arg\min} \{ \int_{-\infty}^{\infty} p(x_3|x_1, x_2) \tag{3}$$
$$f(x_1, x_2, x_3) dx_3 - f(x_1, x_2, x'_3) \}^2,$$

**Algorithm 1** Estimate $\psi'(\cdot)$ in single missing. Assume that the dimensionality of data is 3 and the value missing happens only at the third dimensions.

---

**Require:** $D_{train} = \{(\boldsymbol{x}, y) | \boldsymbol{x} = (x_1, x_2, x_3)$ s.t., $a < x_1, x_2, x_3 < b\}$
**Require:** The number of division $d$
  bandwidth $\leftarrow (b - a)/d$
  **for** $i \leftarrow 0$ to bandwidth $- 1$ **do**
    **for** $j \leftarrow 0$ to bandwidth $- 1$ **do**
      count $\leftarrow 0$
      $y_{\text{sum}} \leftarrow 0$
      **for** $(\boldsymbol{x}^t, y^t) \in D_{train}$ **do**
        **if** $\boldsymbol{x}_1^t$ in $(x_{1i}, x_{1(i+1)}]$ and $(x_{2j}, x_{2(j+1)}]$ **then**
          count $\leftarrow$ count $+ 1$
          $y_{\text{sum}} \leftarrow y_{\text{sum}} + y^t$
        **end if**
        $y_{\text{avg}} \leftarrow y_{\text{sum}}/$count
      **end for**
      $t' \leftarrow \arg\min(y_{\text{avg}} - y^t)^2$
      $x_3 \leftarrow x_3^{t'}$ at $(x_{1i}, x_{1(i+1)}]$ and $(x_{2j}, x_{2(j+1)}]$
    **end for**
  **end for**

---

$$\psi'_2, \psi'_3 = \arg\min_{x'_2, x'_3}\{\int_{-\infty}^{\infty}\int_{-\infty}^{\infty} p(x_2, x_3 | x_1) \tag{4}$$
$$f(x_1, x_2, x_3)dx_2 dx_3 - f(x_1, x'_2, x'_3)\}^2.$$

In order to find $x_2$ and $x_3$ that satisfy Eq. (2) - (4), we first estimate $x_2$ and after that estimate $x_3$ step by step. Thus, we can extend the estimation method that is for single missing to multiple missing. The extended method is described in Algorithm 2 and the more detail explanation of this algorithm is in Appendix B.

## 4. Experiments

In computational experiments, we employ the following benchmark functions:

$\boldsymbol{f_1}$ **(Sphere function):** $\quad f(\boldsymbol{x}) = \sum_{k=1}^{n} x_k^2 \quad (-5 < x_k < 5)$

If we suppose that $n = 3$, $p(x_1, x_2, x_3) = p(x_1)p(x_2)p(x_3)$ and $p(x_1) = p(x_2) = p(x_3) = \frac{1}{10}$ (i.e., a uniform distribution), then the optimal substitution value in the ideal situation is obtained from Eq. (1). The optimal values are described at Appendix C.

$\boldsymbol{f_2}$: $\quad f(\boldsymbol{x}) = (x_1 - x_2 - x_3)^2, \quad (-5 < x_1, x_2, x_3 < 5)$

If we suppose that $n = 3$, $p(x_1, x_2, x_3) = p(x_1)p(x_2)p(x_3)$ and $p(x_1) = p(x_2) = p(x_3) = \frac{1}{10}$ (i.e., a uniform distribution), the optimal substitution value in the ideal situation is

**Algorithm 2** Estimate optimal values in multiple missing. Assume that the dimensionality of data is 3 and the value missing happens at the second and third dimensions.

---

**Require:** $D_{train} = \{(\boldsymbol{x}, y) | \boldsymbol{x} = (x_1, x_2, x_3)$ s.t., $a < x_1, x_2, x_3 < b\}$
**Require:** $\psi'(\cdot)$ in single missing by Algorithm 1
**Require:** The number of division $d$
  bandwidth $\leftarrow (b - a)/d$
  **for** $i \leftarrow 0$ to bandwidth $- 1$ **do**
    count $\leftarrow 0$
    $\alpha_{\text{sum}} \leftarrow 0$
    **for** $(\boldsymbol{x}^t, y^t) \in D_{train}$ **do**
      **if** $x_1^t$ in $(x_{1i}, x_{1(i+1)}]$ **then**
        count $\leftarrow$ count $+ 1$
        $\alpha_{\text{sum}} \leftarrow \alpha_{\text{sum}} + x_3^t$
      **end if**
      $\alpha_{\text{avg}}[i] \leftarrow \alpha_{\text{sum}}/$count
    **end for**
  **end for**
  **if** missing on the second and third dimensions **then**
    $i \leftarrow$ index s.t., $x_1$ in $(x_{1i}, x_{1(i+1)}]$
    $x_2 \leftarrow \psi'_2(x_1, \alpha_{\text{avg}}[i])$
    $x_3 \leftarrow \psi'_3(x_1, x_2)$
  **else if** missing on the second dimension **then**
    $x_2 \leftarrow \psi'_2(x_1, x_3)$
  **else if** missing on the third dimension **then**
    $x_3 \leftarrow \psi'_3(x_1, x_2)$
  **end if**

---

obtained from Eq. (1). The optimal values are also described at Appendix D.

The number $d$ that divides domains of non-missing dimensionality is obtained by the following equation:

$$d = \sqrt[n]{N_{all}}, \tag{5}$$

where $n$ and $N_{all}$ mean the dimensionality and the number of data in $D_{train}$, respectively. In this paper, three-dimensional problems are used for the experiments. The number of $D_{train}$ is 10000, so the number of $d$ is

$$d = \sqrt[3]{10000} = 21.544... \simeq 22,$$

obtained by Eq. (5). Every element of the data is drawn from a uniform random distribution with the domain $(-5, 5)$.

A neural network is employed to model the benchmark functions. The neural network is trained with the following settings: The number of epochs is 1000 and the size of a mini-batch is 32. The number of layers in the neural network is set to three and the number of hidden units is specified as 50. The sigmoid function is used as an activation function for each layer and each unit. Adam algorithm (Kingma & Ba, 2015) is used as the optimizer that computes adaptive learning rates for updating the weights of the networks.

In order to show the effectiveness of the estimation method, we compare the prediction errors of models among several substitution ways for missing values. Substitution probability in the training phase and missing probability in the test phase are set as $p_{sub}, p_{mis} \in \{0.00, 0.25, 0.50, 0.75, 0.90\}$, respectively. When substituting and missing would happen, the values are replaced to $\psi'$ estimated by Algorithms 1 and 2 according to non-missing values.

## 5. Results

The results of the prediction errors when training by using the substitution values with probability $p_{sub}$ are shown in Figs. 1 and 2. The horizontal axes in the figures represent $p_{mis}$. At the setting $p_{sub} = 0.00$, the models are trained without IVS training. In Figs. 1 and 2, the test missing probabilities are changed at the interval of 0.1 for each experimental setting. The test errors in $f_1$ and $f_2$ are compared among five types of the substitution methods. The solid line and the colored areas represent the average and the variance of the test error using the model trained with IVS training, respectively. "Zero" and "Five" set the fixed value 0.0 and 5.0 to the missing features. "Theory" and "Theory random" are set to the substitution value as described in Sec. 2 with the temporary value $\alpha = 0$ (see Appendix B). When the features are missing simultaneously, both of them are imputed according to the substitution methods. The difference between them is that "Theory" indicates the two substitution values (positive and negative) that give preference to the positive one. On the other hand, "Theory random" substitutes the two values randomly. "Estimation" substitutes the value estimated by the Algorithms 1 and 2.

The performance of "Estimation" is as good as "Theory" and "Theory random" for all settings. Therefore, it is shown that the estimated substitution values that obtained from our proposed method work effectively for those data that include missing values regardless of multiple missing even though the target function $f$ is unknown. Moreover, comparing the $y$-axis of (a)-(e) in each setting, it is noteworthy that IVS training allows the models to become more robustness even though any value is employed as the substitution.

In respect of the substitution probability $p_{sub}$, it was found that the proposed method can obtain some effect regardless of the frequency of IVS Training. However, the optimal substitution probability will depend on the test missing probability $p_{mis}$.

For the function $f_2$, the substitution value "Zero" was as good as "Theory" and "Theory random". On the other hand, the setting "Zero" has no effect for the function $f_1$. The policy of assigning 0 is not effective for all functions. It seems reasonable to conclude that the estimation method can be employed for any regression problems in order to obtain a robust model.

## 6. Conclusions

In this research, we extended the estimation method of the optimal substitution value that can consider multiple missing in the IVS training. As the results of numerical experiments, it was shown that the validity of the robust model against the loss for unknown data that contain missing values by estimating the optimal substitution values. For future work, we will conduct experiments with a biased-distribution data, and make use of the findings of this research for handling missing values in other noisy experimental settings.

## Acknowledgement

This work was partially funded by the Tateishi Science and Technology Foundation under the project number 2196102.

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

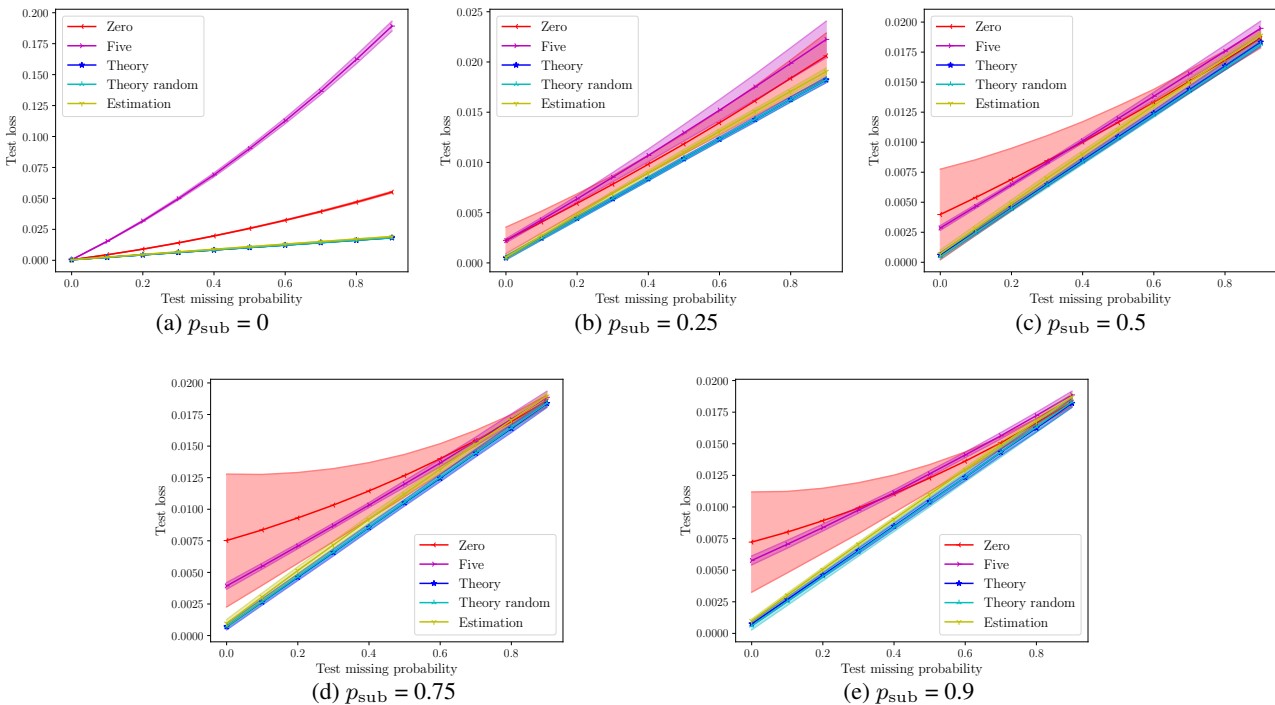

*Figure 1.* Test error maps on $f_1$ (Sphere)

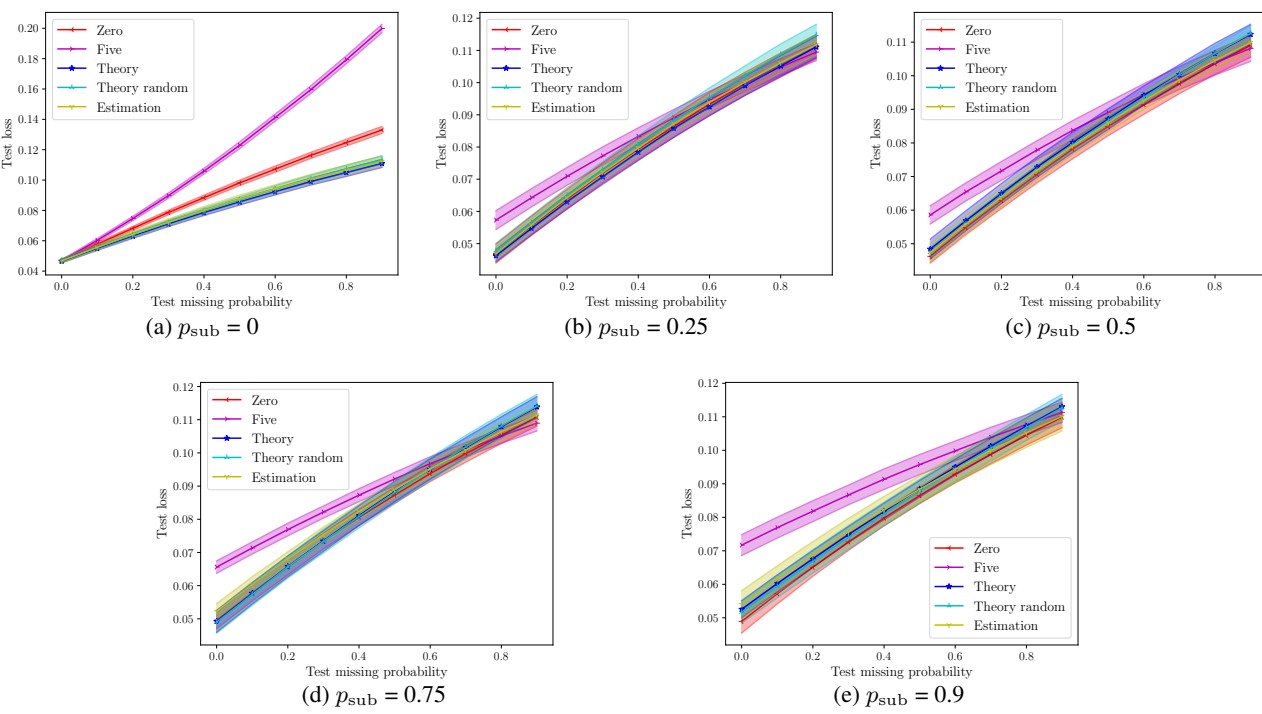

*Figure 2.* Test error maps on $f_2$

## A. Analysis on the Optimal Values with the Target Function

The expected error of the trained model for test data is mathematically investigated. The mathematical investigation reveals that naive substitutions such as an average and a zero do not lead to a good trained model with a high prediction performance for unseen data. It should be noted that the mathematically appropriate substitution value can be obtained only in a ideal situation where the target function is known and which feature will be missing in the prediction phase. Thus, the mathematically appropriate substitution value is used only for the reference in the computational experiments.

Let us denote the $n$ feature variables as an $n$-dimensional random variable vector $\vec{X} = (X_1, X_2, \ldots, X_n)$. We also consider an $n$-dimensional random variable vector $\vec{R} = (R_1, R_2, \ldots, R_n)$, where each element of the vector represents whether the corresponding feature is observed or missing as follows:

$$R_i = \begin{cases} 1, & \text{if } X_i \text{ is observed,} \\ 0, & \text{otherwise (i.e., } X_i \text{ is missing).} \end{cases} \quad (6)$$

Now let us define a new random variable as follows:

$$X_i' = \begin{cases} X_i, & \text{if the } i\text{-th feature value is observed,} \\ ?, & \text{if it is missing.} \end{cases} \quad (7)$$

Then, we can define $\phi : X \times R \longrightarrow X'$, where $\phi$ is a bijective function.

When we consider the modeling problem with missing data using a joint probability distribution on the universe of discourse $(X_1, \ldots, X_n, R_1, \ldots, R_n)$, the joint probability function $p(\boldsymbol{x}, \boldsymbol{r})$ is defined as follows:

$$p(\boldsymbol{x}, \boldsymbol{r}) = p(\boldsymbol{x}|\boldsymbol{r})p(\boldsymbol{r}) = p(\boldsymbol{r}|\boldsymbol{x})p(\boldsymbol{x}), \quad (8)$$

where $p(\boldsymbol{x}) = p(x_1, \ldots, x_n)$ is the joint probability density function of $X_1, \ldots, X_n$, and $p(\boldsymbol{r}|\boldsymbol{x})$ is the probability function which represents whether $x_i$ is observed or not for $X = \boldsymbol{x}$.

Secondly, we define a substituting operation for missing elements of the feature values. Let a mapping be $\psi : \mathbb{R}_?^n \to \mathbb{R}^n$ where $\mathbb{R}_?^n$ is $\{\boldsymbol{x}' = (x_1', \cdots, x_n')|x_i' \in \mathbb{R} \cup \{?\}\}$. Then the substituted data follow $\boldsymbol{x}^* = \psi(\boldsymbol{x}') = \psi(\phi(\boldsymbol{x}, \boldsymbol{r}))(\triangleq \psi^{\boldsymbol{r}}(\boldsymbol{x}))$. Furthermore, when we put $\psi^{\boldsymbol{r}}(\boldsymbol{x}) = \psi_1^{\boldsymbol{r}} \times \cdots \times \psi_n^{\boldsymbol{r}}(\boldsymbol{x}) = (\psi_1^{\boldsymbol{r}}(\boldsymbol{x}), \ldots, \psi_n^{\boldsymbol{r}}(\boldsymbol{x}))$, we can obtain

$$\psi_i(\phi(\boldsymbol{x}, \boldsymbol{r})) = \begin{cases} x_i, & \text{if } r_i = 1, \\ \psi_i'^{\boldsymbol{r}}(\boldsymbol{x}_{\text{obs}}), & \text{if } r_i = 0, \end{cases} \quad (9)$$

where $\boldsymbol{x}_{\text{obs}}$ is a vector that consists of those observed features.

Next, we discuss the machine learning model and its loss function for a task. For simplicity, let the target function be $f$, and the prediction model be $g$. Without loss of generality, we suppose $f : \mathbb{R}^n \to \mathbb{R}$ and $g : \mathbb{R}^n \to \mathbb{R}$. Moreover, let us define the distance (i.e., error) between $f$ and $g$ for an input vector $\boldsymbol{x}$ as $\delta(f(\boldsymbol{x}), g(\boldsymbol{x}))$, and also let us define a possible vector set for $\boldsymbol{r}$ as $S = \{s_1, \cdots, s_n | \forall i \in \mathbb{N}, s_i \in \{0, 1\}\}$. Then, the expectation of the error $\delta$ between $f$ and $g$ is represented as follows:

$$\mathbb{E}[\delta(f, g)]$$
$$= \sum_{\boldsymbol{s} \in S} \int \cdots \int_{D_X} p(\boldsymbol{x}, \boldsymbol{r} = \boldsymbol{s}) \delta(f(\boldsymbol{x}), g(\psi^{\boldsymbol{s}}(\boldsymbol{x}))) dX$$
$$= \int \cdots \int_{D_X} p(\boldsymbol{x}, \boldsymbol{r} = \boldsymbol{1}) \delta(f(\boldsymbol{x}), g(\boldsymbol{x})) dX \quad (10)$$
$$+ \sum_{\boldsymbol{s} \in S \setminus \{\boldsymbol{1}\}} \int \cdots \int_{D_X} p(\boldsymbol{x}, \boldsymbol{r} = \boldsymbol{s}) \delta(f(\boldsymbol{x}), g(\psi^{\boldsymbol{s}}(\boldsymbol{x}))) dX.$$

Unless otherwise noted, we denote $\int \cdots \int_{D_X} = \int_{D_X}$ for simplifying equations hereafter. In Eq. (10), the first term is the expectation for those input vectors with no missing values, and the second term means the one for those input vectors with missing feature values. Here, when we suppose that the loss is evaluated by $\delta(f, g) = \{f - g\}^2$, then we have the following equation for obtaining the expected loss:

$$\mathbb{E}[\delta(f, g)]$$
$$= \int_{D_X} p(\boldsymbol{x}, \boldsymbol{r} = \boldsymbol{1}) \{f(\boldsymbol{x}) - g(\boldsymbol{x})\}^2 dX \quad (11)$$
$$+ \sum_{\boldsymbol{s} \in S \setminus \{\boldsymbol{1}\}} \int_{D_X} p(\boldsymbol{x}, \boldsymbol{r} = \boldsymbol{s}) \{f(\boldsymbol{x}) - g(\psi^{\boldsymbol{s}}(\boldsymbol{x}))\}^2 dX.$$

Now, we focus only on a single term in the latter part of Eq. (11). In this discussion, we assume that the elements of $\boldsymbol{s}$ are $s_1 = s_2 = \cdots = s_k = 1, s_{k+1} = s_{k+2} = \cdots = s_n = 0$. However, please note that the following discussion holds even if the value of either 1 or 0 appears in arbitrary elements. Let us denote $X_{\text{obs}} = X_1, X_2, \cdots, X_k$, and $X_{\text{mis}} = X_{k+1}, X_{k+2}, \cdots, X_n$. When we suppose $p(\boldsymbol{x}, \boldsymbol{r} = \boldsymbol{s}) = p_{\boldsymbol{s}}(\boldsymbol{x})$, the latter term that satisfies $\boldsymbol{r} = \boldsymbol{s}$ in

Eq. (11) is written as follows:

$$\int_{D_X} p_{\boldsymbol{s}}(\boldsymbol{x})\{f(\boldsymbol{x}) - g(\psi^{\boldsymbol{s}}(\boldsymbol{x}))\}^2 dX$$

$$= \int_{D_X} p_{\boldsymbol{s}}(\boldsymbol{x})f^2(\boldsymbol{x})dX$$

$$- 2\int_{D_{X_{\mathrm{obs}}}} g(\boldsymbol{x}_{\mathrm{obs}}, \psi'^{\boldsymbol{s}}_{k+1}(\boldsymbol{x}_{\mathrm{obs}}), \ldots, \psi'^{\boldsymbol{s}}_n(\boldsymbol{x}_{\mathrm{obs}}))$$

$$\left[\int_{D_{X_{\mathrm{mis}}}} p_{\boldsymbol{s}}(\boldsymbol{x})f(\boldsymbol{x})\, dX_{\mathrm{mis}}\right] dX_{\mathrm{obs}}$$

$$+ \int_{D_{X_{\mathrm{obs}}}} g^2(\boldsymbol{x}_{\mathrm{obs}}, \psi'^{\boldsymbol{s}}_{k+1}(\boldsymbol{x}_{\mathrm{obs}}), \ldots, \psi'^{\boldsymbol{s}}_n(\boldsymbol{x}_{\mathrm{obs}}))$$

$$\left[\int_{D_{X_{\mathrm{mis}}}} p_{\boldsymbol{s}}(\boldsymbol{x})\, dX_{\mathrm{mis}}\right] dX_{\mathrm{obs}}, \qquad (12)$$

where $\int_{D_{X_{\mathrm{mis}}}} p_{\boldsymbol{s}}(\boldsymbol{x})\, dX_{\mathrm{mis}}$ in Eq. (12) can be deformed as follows:

$$\int_{D_{X_{\mathrm{mis}}}} p_{\boldsymbol{s}}(\boldsymbol{x})\, dX_{\mathrm{mis}} = p_{\boldsymbol{s}}(\boldsymbol{x}_{\mathrm{obs}}) \int_{D_{X_{\mathrm{mis}}}} p_{\boldsymbol{s}}(\boldsymbol{x}_{\mathrm{mis}}|\boldsymbol{x}_{\mathrm{obs}})\, dX_{\mathrm{mis}}$$

that denotes a marginal distribution by $X_{\mathrm{mis}}$. By rearranging $\int_{D_{X_{\mathrm{mis}}}} p_{\boldsymbol{s}}(\boldsymbol{x})f(\boldsymbol{x})\, dX_{\mathrm{mis}}$, the following equation is obtained:

$$\int_{D_{X_{\mathrm{mis}}}} p_{\boldsymbol{s}}(\boldsymbol{x})f(\boldsymbol{x})\, dX_{\mathrm{mis}}$$

$$= p_{\boldsymbol{s}}(\boldsymbol{x}_{\mathrm{obs}}) \int_{D_{X_{\mathrm{mis}}}} p_{\boldsymbol{s}}(\boldsymbol{x}_{\mathrm{mis}}|\boldsymbol{x}_{\mathrm{obs}})f(\boldsymbol{x})\, dX_{\mathrm{mis}}$$

$$= p_{\boldsymbol{s}}(\boldsymbol{x}_{\mathrm{obs}})\mathbb{E}_{X_{\mathrm{mis}}}[f(\boldsymbol{x})]. \qquad (13)$$

This represents the expected output value for the missed value. For simplicity, we denote this as $\mathbb{E}_{X_{\mathrm{mis}}}[f(\boldsymbol{x})] = e_s(\boldsymbol{x}_{\mathrm{obs}})$ in the following equations. Based on these discussions, by setting $g'(\boldsymbol{x}_{\mathrm{obs}}) \triangleq g(\boldsymbol{x}_{\mathrm{obs}}, \psi'^{\boldsymbol{s}}_{k+1}(\boldsymbol{x}_{\mathrm{obs}}), \ldots, \psi'^{\boldsymbol{s}}_n(\boldsymbol{x}_{\mathrm{obs}}))$, then we have the following:

$$\int_{D_X} p_{\boldsymbol{s}}(\boldsymbol{x})\{f(\boldsymbol{x}) - g(\psi^{\boldsymbol{s}}(\boldsymbol{x}))\}^2 dX$$

$$= \int_{D_{X_{\mathrm{obs}}}} p_{\boldsymbol{s}}(\boldsymbol{x}_{\mathrm{obs}})\{g'(\boldsymbol{x}_{\mathrm{obs}}) - e_{\boldsymbol{s}}(\boldsymbol{x}_{\mathrm{obs}})\}^2$$

$$- p_{\boldsymbol{s}}(\boldsymbol{x}_{\mathrm{obs}})e_{\boldsymbol{s}}^2(\boldsymbol{x}_{\mathrm{obs}})\, dX_{\mathrm{obs}}$$

$$+ \int_{D_X} p_{\boldsymbol{s}}(\boldsymbol{x})f^2(\boldsymbol{x})dX. \quad (14)$$

If there is only one combination of the observed features and the missing features, that is, if $\boldsymbol{r} = \boldsymbol{s}$, the optimal model $g'$ that minimizes Eq. (14) can be trained from the training using IVS method. Eq. (14) is minimized when $g'(\boldsymbol{x}_{\mathrm{obs}}) = g(\boldsymbol{x}_{\mathrm{obs}}, \psi'^{\boldsymbol{s}}_{k+1}(\boldsymbol{x}_{\mathrm{obs}}), \ldots, \psi'^{\boldsymbol{s}}_n(\boldsymbol{x}_{\mathrm{obs}})) = e_{\boldsymbol{s}}(\boldsymbol{x}_{\mathrm{obs}})$. As it is possible that there is no missing

value in the input vector, it is necessary to minimize Eq. (11), with which Eq. (14) is substituted. Now, we suppose that $f$ can be approximated by $g$, and $g \simeq f$. Then, we obtain the function $\psi'^{\boldsymbol{s}}_{k+1}, \ldots, \psi'^{\boldsymbol{s}}_n$ satisfying $g(\boldsymbol{x}_{\mathrm{obs}}, \psi'^{\boldsymbol{s}}_{k+1}(\boldsymbol{x}_{\mathrm{obs}}), \ldots, \psi'^{\boldsymbol{s}}_n(\boldsymbol{x}_{\mathrm{obs}})) = e_{\boldsymbol{s}}(\boldsymbol{x}_{\mathrm{obs}})$. This leads to the substitution value for minimizing the expectation of error in the case where the above missing observations could occur.

Finally, we discuss this mathematical analysis in more detail by tackling to a concrete example. Let us assume that $R_1, R_2, \cdots, R_n$, and $X$ are independent, that is, $p(\boldsymbol{x}, \boldsymbol{r}) = p(\boldsymbol{x})p(r_1)p(r_2)\ldots p(r_n)$. Moreover, we also suppose $p(r_1 = 1) = \cdots = p(r_k = 1) = 1.0$, $p(r_{k+1} = 0) = \cdots = p(r_n = 0) = p_{\mathrm{mis}}$. The expected error under these settings is obtained from Eq. (11) as follows:

$$\mathbb{E}[\delta(f,g)] = (1 - p_{\mathrm{mis}}) \int_{D_{X_{\mathrm{obs}}}} p(\boldsymbol{x})\{f(\boldsymbol{x}) - g(\boldsymbol{x})\}^2 dX_{\mathrm{obs}}$$

$$+ p_{\mathrm{mis}} \int_{D_{X_{\mathrm{mis}}}} p(\boldsymbol{x})\{f(\boldsymbol{x}) - g(\psi^{\boldsymbol{s}}(\boldsymbol{x}))\}^2 dX_{\mathrm{mis}}$$

and from Eq. (14), we obtain

$$\mathbb{E}[\delta(f,g)] = (1 - p_{\mathrm{mis}}) \int_{D_X} p(\boldsymbol{x})\{f(\boldsymbol{x}) - g(\boldsymbol{x})\}^2 dX$$

$$+ p_{\mathrm{mis}} \int_{D_{X_{\mathrm{obs}}}} p(\boldsymbol{x}_{\mathrm{obs}})\{g'(\boldsymbol{x}_{\mathrm{obs}}) - e(\boldsymbol{x}_{\mathrm{obs}})\}^2$$

$$- p(\boldsymbol{x}_{\mathrm{obs}})e^2(\boldsymbol{x}_{\mathrm{obs}})\, dX_{\mathrm{obs}}$$

$$+ p_{\mathrm{mis}} \int_{D_X} p(\boldsymbol{x})f^2(\boldsymbol{x})dX,$$

where $e(\boldsymbol{x}_{\mathrm{obs}}) = \mathbb{E}_{X_{\mathrm{mis}}}[f(\boldsymbol{x})] = \int_{-\infty}^{\infty} p(\boldsymbol{x}_{\mathrm{mis}}|\boldsymbol{x}_{\mathrm{obs}})\, f(\boldsymbol{x})\, dX_{\mathrm{mis}}$ (see Eq. (13) ). Now, when $g \simeq f$, the expected error is minimized if $\psi'$ satisfies the following equation:

$$\psi'_{\boldsymbol{x}_{\mathrm{mis}}}(\boldsymbol{x}_{\mathrm{obs}})$$

$$= \arg\min_{\boldsymbol{x}'_{\mathrm{mis}}} \{\int_{D_{X_{\mathrm{mis}}}} p(\boldsymbol{x}_{\mathrm{mis}}|\boldsymbol{x}_{\mathrm{obs}})f(\boldsymbol{x}_{\mathrm{obs}}, \boldsymbol{x}_{\mathrm{mis}})\, dX_{\mathrm{mis}}$$

$$- f(\boldsymbol{x}_{\mathrm{obs}}, \boldsymbol{x}'_{\mathrm{mis}})\}^2. \qquad (15)$$

## B. Optimal Temporary Value for the Proposed Method

When $\psi'$ is a multi-valued function, we suppose that one of the solutions is selected according to pre-defined rules (i.e., $\psi'$ becomes a one-to-one correspondence function).

We can use Eq. (2) in order to obtain the optimal substitution values on the second dimension $x_2$. However, since the value on the third dimension is missing, let us put $x_3 = \alpha$ temporary. Then, we can obtain the optimal value $x_2$ as

follows:

$$\psi_2'(x_1, \alpha) = \arg\min_{x_2'}\{\int_{-\infty}^{\infty} p(x_2|x_1, \alpha)f(x_1, x_2, \alpha)dx_2$$
$$-f(x_1, x_2', \alpha)\}^2. \qquad (16)$$

By using Eq.(16), the optimal values of the third dimansion are written as follows:

$$\psi_3'(x_1, \psi_2'(x_1, \alpha))$$
$$= \arg\min_{x_3'}\{\int_{-\infty}^{\infty} p(x_3|x_1, \psi_2'(x_1, \alpha))$$
$$f(x_1, \psi_2'(x_1, \alpha), x_3)dx_3$$
$$-f(x_1, \psi_2'(x_1, \alpha), x_3')\}^2$$
$$= \arg\min_{x_3'}\{\int_{-\infty}^{\infty} p(x_3|x_1)f(x_1, \psi_2'(x_1, \alpha), x_3)dx_3$$
$$-f(x_1, \psi_2'(x_1, \alpha), x_3')\}^2, \qquad (17)$$

where $p(x_1 = x_1' \cap \psi_2'(x_1, \alpha) = \psi_2'(x_1', \alpha)) = p(x_1 = x_1')$ because $\psi'$ is a one-to-one correspondence function.

The term $\int_{-\infty}^{\infty} p(x_3|x_1)f(x_1, \psi_2'(x_1, \alpha), x_3)dx_3$ in Eq. (17) is the expected value of $f$ for $x_3$ when $x_2$ is set to $\psi_2'(x_1, \alpha)$. On the other hand, from Eq. (4), the expected value of $f(\cdot)$ is $\int_{-\infty}^{\infty}\int_{-\infty}^{\infty} p(x_2, x_3|x_1)f(x_1, x_2, x_3)dx_2dx_3$ when $x_2$ and $x_3$ are simultaneously missing. Therefore, $\alpha$ that equalize the expected value in Eq. (17) to the one of Eq. (4) can be obtained by the following equation:

$$\int_{-\infty}^{\infty} p(x_3|x_1)f(x_1, \psi_2'(x_1, \alpha), x_3)dx_3$$
$$= \int_{-\infty}^{\infty}\int_{-\infty}^{\infty} p(x_2, x_3|x_1)f(x_1, x_2, x_3)dx_2dx_3. (18)$$

The right side of the Eq. (18) can be rearranged by Bayes' theorem as follows:

$$\int_{-\infty}^{\infty}\int_{-\infty}^{\infty} p(x_2, x_3|x_1)f(x_1, x_2, x_3)dx_2dx_3$$
$$= \int_{-\infty}^{\infty}\int_{-\infty}^{\infty} p(x_2|x_1, x_3)p(x_3|x_1)f(x_1, x_2, x_3)dx_2dx_3$$
$$= \int_{-\infty}^{\infty} p(x_3|x_1)$$
$$\{\int_{-\infty}^{\infty} p(x_2|x_1, x_3)f(x_1, x_2, x_3)dx_2\}dx_3. (19)$$

Now, from Eq. (2), the term $\int_{-\infty}^{\infty} p(x_2|x_1, x_3)f(x_1, x_2, x_3)dx_2$ is the expected value of $f(\cdot)$ for $x_2$. In the case of continuous function $f(\cdot)$, there is at least one $x_2'$ satisfying $f(x_1, x_2', x_3) - \int_{-\infty}^{\infty} p(x_2|x_1, x_3)f(x_1, x_2, x_3)dx_2 = 0$ that can minimizes the right part of Eq. (2). Therefore, Eq. (2) can be deformed as follows:

$$f(x_1, \psi_2'(x_1, x_3), x_3)$$
$$= \int_{-\infty}^{\infty} p(x_2|x_1, x_3)f(x_1, x_2, x_3)dx_2. \qquad (20)$$

By substituting Eq. (20) into Eq. (19), we can obtain the following equation:

$$\int_{-\infty}^{\infty}\int_{-\infty}^{\infty} p(x_2, x_3|x_1)f(x_1, x_2, x_3)dx_2dx_3$$
$$= \int_{-\infty}^{\infty} p(x_3|x_1)\{\int_{-\infty}^{\infty} p(x_2|x_1, x_3)f(x_1, x_2, x_3)dx_2\}dx_3$$
$$= \int_{-\infty}^{\infty} p(x_3|x_1)f(x_1, \psi_2'(x_1, x_3), x_3)dx_3. \qquad (21)$$

Therefore, the value of $\alpha$ that holds Eq. (18) is written as follows:

$$\int_{-\infty}^{\infty} p(x_3|x_1)f(x_1, \psi_2'(x_1, \alpha), x_3)dx_3$$
$$= \int_{-\infty}^{\infty} p(x_3|x_1)f(x_1, \psi_2'(x_1, x_3), x_3)dx_3$$
$$\Leftarrow \int_{-\infty}^{\infty} p(x_3|x_1)\alpha dx_3 = \int_{-\infty}^{\infty} p(x_3|x_1)x_3dx_3$$
$$\Leftrightarrow \alpha = \int_{-\infty}^{\infty} p(x_3|x_1)x_3dx_3, (22)$$

where the right side of Eq. (22) means a conditional expectation of the random variable $x_3$ for $x_1$. When we temporarily substitute the optimal value to $\alpha$, the relational expression Eq. (4) can be satisfied.

$$(\psi_2'(x_1, \alpha), \psi_3'(x_1, \psi_2'(x_1, \alpha)))$$
$$\in \arg\min_{x_2', x_3'}\{\int_{-\infty}^{\infty}\int_{-\infty}^{\infty} p(x_2, x_3|x_1)f(x_1, x_2, x_3)dx_2dx_3$$
$$-f(x_1, x_2', x_3')\}^2. \qquad (23)$$

## C. Optimal Values for $f_1$

$$\psi_2'(x_1, x_3)$$
$$= \arg\min_{x_2}\{\int_{-5}^{5} \frac{1}{10}(x_1^2 + x_2^2 + x_3^2)dx_2 - (x_1^2 + x_2^2 + x_3^2)\}^2$$
$$= \pm\frac{5}{\sqrt{3}},$$

$$\psi_3'(x_1, x_2)$$
$$= \arg\min_{x_3}\{\int_{-5}^{5} \frac{1}{10}(x_1^2 + x_2^2 + x_3^2)dx_3 - (x_1^2 + x_2^2 + x_3^2)\}^2$$
$$= \pm\frac{5}{\sqrt{3}},$$

$$\psi_{2,3}'(x_1)$$
$$= \arg\min_{x_2, x_3}\{\int_{-5}^{5}\int_{-5}^{5} \frac{1}{100}(x_1^2 + x_2^2 + x_3^2)\,dx_2dx_3$$
$$-(x_1^2 + x_2^2 + x_3^2)\}^2$$
$$\Leftrightarrow x_2^2 + x_3^2 = 50/3.$$

## D. Optimal Values for $f_2$

$$\psi_2'(x_1, x_3) = \arg\min_{x_2}\{\int_{-5}^{5} \frac{1}{10}(x_1 - x_2 - x_3)^2 \, dx_2$$
$$-(x_1 - x_2 - x_3)^2\}^2$$
$$= (x_1 - x_3) \pm \sqrt{(x_1 - x_3)^2 + \frac{25}{3}},$$

$$\psi_3'(x_1, x_2) = \arg\min_{x_3}\{\int_{-5}^{5} \frac{1}{10}(x_1 - x_2 - x_3)^2 \, dx_3$$
$$-(x_1 - x_2 - x_3)^2\}^2$$
$$= (x_1 - x_2) \pm \sqrt{(x_1 - x_2)^2 + \frac{25}{3}},$$

$$\psi_{2,3}'(x_1) = \arg\min_{x_2,x_3}\{\int_{-5}^{5}\int_{-5}^{5} \frac{1}{100}(x_1 - x_2 - x_3)^2 \, dx_2 dx_3$$
$$-(x_1 - x_2^2 - x_3)^2\}^2$$
$$\Leftrightarrow \quad x_2 + x_3 = x_1 \pm \sqrt{x_1^2 + 50/3}.$$