# OpenReview forum: "A Study on Intentional-Value-Substitution Training for Regression with Incomplete Information"
_ICML.cc/2020/Workshop/Artemiss — ICML Artemiss 2020_

### Official Review · AnonReviewer1 · 2020-06-23
**Interesting topic, but the method applies to very restrictive settings**

**Confidence:** 3
**Rating:** 6

**Review:**

**Summary**:
The authors consider a regression problem where values are missing in the test set but not in the training set. They consider a previously published method called Intentional-Value-Substitution (IVS). At train time, this method replaces known values from the training set by new values chosen so as to minimise the difference between the conditional expectation of the target function with respect to p(x_mis|x_obs) and the target function evaluated at these new values. IVS thus needs knowledge of the true target function. Previous work has proposed an algorithm to overcome this limitation when at most one variable has missing entries in the test set. In this work, the authors extend this algorithm to the case of two variables with missing entriers in the test set.

**Pros**:
* The problem of supervised learning with missing values is an interesting one, which has received little attention so far.

**Cons**:
* The method proposed applies to quite restrictive settings: no missing values in the train set, problems with at most 2 variables with missing entries in the test set, at least one variable fully observed, all variables independent.
* More insights to explain the results of the experiments would be useful. See questions.

**Questions**:
* It would be nice to clearly state how the test points are handled. At test time, the missing entries are imputed with the values computed at train time according to the region (x_1i, x_1(i+1)] it falls into?
* Figure 2 shows that substituting values from the train set by 0 performs as well as IVS. But it is not the case in Figure 1. Comments on that would be interesting, because it would highlight when the simple procedure of replacing values by 0 is sufficient and when it is not.
*  It seems that the probability of substitution (say between 0.25 and 0.9) in the train set has little impact on the performance (for Theory, Theory random and Estimation at least). This is surprising to me. How high should the probability of substitution be to degrade performances?
* To obtain eq. 19, you use an equality, but how is this equality obtained? \psi’_2(x_1, x_3) is defined as an argmin, but I don’t see why it would imply the equality used to obtain eq.19

**Remarks**:
* The difference between Theory and Theory random is not clear to me.

---

### Decision · Program_Chairs · 2020-07-02

**Decision:**

Accept

**Comment:**

We are very happy to inform you that your paper has been accepted for the Artemiss workshop. We will contact you soon to inform you about the details concerning the format of your presentation at the workshop, and the camera-ready version deadline. Please take into account the referee's comments to write the camera-ready version.